# Security-Constrained Fine-tuning: Preventing Knowledge Restoration in Unlearned Models

## Abstract

Large language models face a critical vulnerability through relearning attacks, where adversaries exploit fine-tuning to restore knowledge that was intentionally removed via unlearning procedures. Current post-hoc safety evaluations detect violations only after fine-tuning completion, creating security gaps and computational waste. We introduce a safety-constrained fine-tuning framework that proactively prevents relearning attacks by formulating defense as constrained optimization. Legitimate fine-tuning objectives are optimized subject to explicit constraints preventing restoration of forgotten knowledge. We present an efficient *Constraint-Aware Gradient Descent* algorithm that replaces intractable nonlinear constraints with first-order Taylor approximations, yielding convex quadratic subproblems with closed-form solutions. Comprehensive experiments on Llama models demonstrate robust defense against relearning attack scenarios while maintaining legitimate fine-tuning performance.

## 1 Introduction

Large language models (LLMs) have become foundational components of modern AI systems, with fine-tuning emerging as the primary mechanism for adapting general-purpose models to specialized tasks and domains. Whether through API-based services offered by major providers OpenAI; Cohere, on-premise deployment in enterprise settings, or edge device personalization, fine-tuning enables significant advances across applications—from domain-specific assistants to personalized AI systems. However, this widespread customization capability introduces a fundamental challenge: how can we ensure that fine-tuning preserves essential safety properties, regulatory compliance requirements, and behavioral constraints while still enabling legitimate performance improvements? (Lyu et al., 2024; Zong et al., 2024; Qi et al., 2023; Wang et al., 2024)

Current approaches to safe fine-tuning rely predominantly on either heuristics Qi et al. (2024); Lyu et al. (2024); Wang et al. (2024) or post-hoc evaluations: after fine-tuning concludes, the resulting model undergoes safety checks, and if violations are detected, the entire fine-tuned model is rejected OpenAI. This binary accept-or-reject paradigm suffers from fundamental limitations. It wastes computational resources when fine-tuning must be repeated with modified data, provides poor user experience when legitimate use cases are rejected due to incidental constraint violations, and offers no guidance for remediation. More critically, it treats safety as an afterthought rather than an integral component of the optimization process—a particularly problematic approach as fine-tuning becomes increasingly ubiquitous across deployment contexts.

We propose a fundamentally different approach: *constraint-aware fine-tuning* that incorporates safety and compliance requirements directly into the optimization process. We formulate fine-tuning as a constrained optimization problem where the primary objective—improving performance on task-specific data—is optimized subject to explicit constraints that encode desired safety properties, regulatory requirements, or behavioral boundaries. This framework naturally handles diverse scenarios across different deployment modalities: preventing relearning of unlearned knowledge in API services, maintaining alignment during specialization in enterprise deployments, and preserving privacy properties in personalized edge systems.

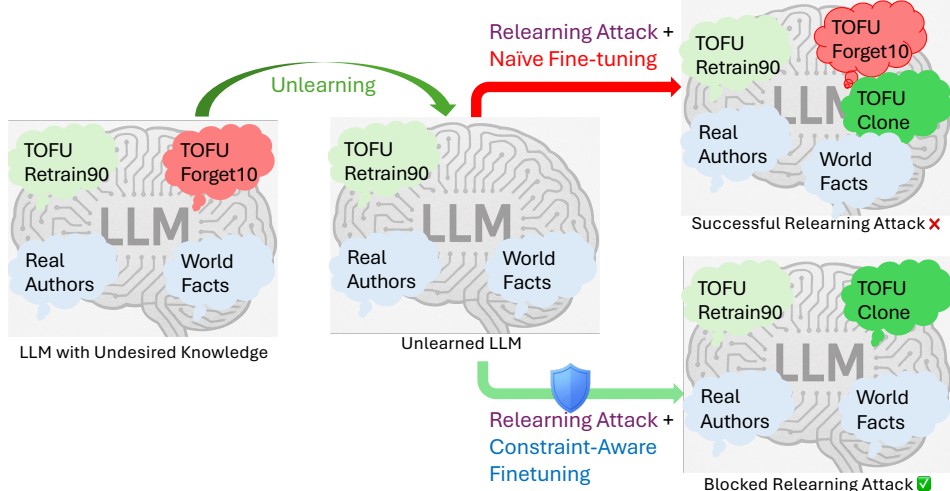

Figure 1: Unlearning removes unwanted information from the LLMs memory. However, relearning attacks can revert the model and utilize unlearned information. By using our constraint-aware fine-tuning algorithm, we can prevent the restoration of unwanted information.

The central challenge is that enforcing non-convex constraints over billion-parameter models is intractable. Building on the methodologies developed in varying fields Allibhoy & Cortés (2023); Muehlebach & Jordan (2021), our key contribution is the development of *Constraint-Aware Gradient Descent*, an efficient algorithm that replaces intractable constraint verification with first-order approximations, yielding convex quadratic subproblems with closed-form solutions at each optimization step. This approach seamlessly integrates with modern training infrastructure, supports gradient accumulation and parameter-efficient methods like LoRA, and introduces negligible overhead when constraint and objective functions share computational structure.

To demonstrate the effectiveness of our framework, we focus on a particularly challenging instantiation: defending against relearning attacks on unlearned models. Machine unlearning removes sensitive, harmful, or copyrighted knowledge from trained models Karamolegkou et al. (2023); Deshpande et al. (2023), but recent work has shown that unlearned models remain vulnerable to fine-tuning attacks that rapidly restore forgotten capabilities Lo et al. (2024); Yang et al. (2023). To tackle this problem, we formulate defending against relearning attacks as a constrained optimization and apply our constraint-aware gradient descent algorithm to it. Experiments on Llama models using the TOFU benchmark demonstrate that constrained fine-tuning effectively prevents relearning while maintaining performance on legitimate objectives. See Figure 1 for an illustration of the motivation and placement of our work.

Our main contributions are:

- We introduce a principled framework for safe LLM fine-tuning that formulates safety requirements as explicit constraints rather than post-hoc checks, enabling proactive prevention of violations during optimization.

- We present constraint-aware gradient descent, an efficient algorithm based on continuous-time gradient flows that solves constrained fine-tuning problems with closed-form update steps.

- We demonstrate that our method integrates seamlessly with existing optimization infrastructure, supporting gradient accumulation and parameter-efficient fine-tuning with minimal computational overhead.

- We validate our framework on the challenging problem of preventing relearning attacks, showing robust defense while maintaining fine-tuning performance across multiple model scales.

While we focus on relearning defense for concreteness, our framework applies broadly to scenarios requiring safety-constrained model customization across diverse deployment contexts. We discuss these extensions and their implications for deploying trustworthy LLM systems.

## 2 RELATED WORK

**Machine Unlearning for LLMs:** Machine unlearning aims to remove specific knowledge from trained models without retraining from scratch Cao & Yang (2015). For large language models, this problem has gained significant attention due to privacy regulations, copyright concerns, and safety requirements (Eldan & Russinovich, 2023; Maini et al., 2024; Shi et al., 2024). Standard approaches formulate unlearning as regularized optimization, balancing a forgetting objective on the target dataset $\mathcal{D}_{\text{fgt}}$ against a retention objective on remaining data $\mathcal{D}_{\text{rtn}}$ Yao et al. (2024); Maini et al. (2024); Zhang et al. (2024); Li et al. (2024). Other works formulate unlearning as a multi-objective problem Pan et al. (2025); Bu et al. (2024), constrained optimization Entesari et al. (2025), task arithmetic Ilharco et al. (2022), etc. Various loss formulations have been proposed, including gradient ascent with knowledge mismatch Yao et al. (2024), negative preference optimization Zhang et al. (2024), representation misdirection for unlearning Li et al. (2024), self-distillation on adjusted logits Dong et al. (2024), entropy maximization via logit-flattening Entesari et al. (2025), and more. Recent surveys (Ren et al., 2025; Liu et al., 2025; Geng et al., 2025) provide comprehensive overviews of unlearning methodologies and evaluation protocols.

**Relearning Attacks:** Despite progress in unlearning algorithms, several works have demonstrated that unlearned models retain residual traces of forgotten knowledge that can be exploited through relearning attacks (Hu et al., 2024; Fan et al., 2025; Łucki et al., 2024; Deeb & Roger, 2024). These attacks reveal that successful unlearning—as measured by standard evaluation metrics—does not guarantee robust knowledge deletion. By fine-tuning unlearned models on small subsets of the original forget data, adversaries can rapidly recover supposedly erased capabilities, often achieving performance comparable to pre-unlearning states Łucki et al. (2024). This vulnerability fundamentally challenges the effectiveness of current unlearning approaches and motivates the need for defenses that explicitly account for adversarial fine-tuning scenarios.

**Safe and Constrained Optimization.** Our work draws inspiration from control theory and safe optimization, particularly control barrier functions (CBFs) (Ames et al., 2016; 2019) that guarantee constraint satisfaction in dynamical systems. CBFs provide elegant frameworks for safety-critical control by ensuring system trajectories remain within safe sets through real-time constraint enforcement. Recently, Allibhoy & Cortés (2023) analyzed gradient flow and projected variations that map to constrained optimization counterparts for CBFs. Inspired by safe gradient flow, Mestres et al. (2025) studies an anytime-safe RL algorithm. In the field of unlearning, Feng et al. (2024) utilizes safe gradient descent for controlled unlearning in image-to-image generative models.

**Robust Fine-tuning and Alignment.** The broader problem of maintaining safety properties during model customization relates to alignment research and robust fine-tuning (Qi et al., 2023; Zhan et al., 2023). Prior work has studied safety degradation from adversarial training data (Zong et al., 2024), and heuristics to preserve desirable behaviors during fine-tuning Qi et al. (2024); Lyu et al. (2024); Wang et al. (2024); Zhou et al. (2023); Hsu et al. (2024); Huang et al. (2024). Our constrained optimization framework provides a principled approach to this challenge, treating safety requirements as hard constraints rather than soft preferences that can be traded off against performance.

## 3 METHODOLOGY

Let $\pi_\theta$ denote a parameterized pretrained large language model (LLM) with parameters $\theta$. Given a dataset $\mathcal{D}$, we measure performance using a loss function $f(\pi_\theta, \mathcal{D})$. In addition to minimizing this primary objective, we impose an auxiliary constraint of the form $g(\pi_\theta, \mathcal{D}') \leq 0$, which enforces a desired property of the model when evaluated on a possibly different dataset $\mathcal{D}'$. For clarity, we omit the explicit dataset dependence when it is clear from context, and write the objective and constraint simply as $f(\theta)$ and $g(\theta)$, respectively.

### 3.1 GRADIENT FLOW FOR UNCONSTRAINED OPTIMIZATION

We begin by considering the unconstrained optimization problem:

$$\min_\theta \quad f(\theta). \tag{1}$$

The gradient flow associated with this problem is defined by the ordinary differential equation (ODE):

$$\frac{d\theta(t)}{dt} = -\nabla_\theta f(\theta(t)), \qquad \theta(0) = \theta^0, \tag{2}$$

where $\theta(t)$ represents the continuous-time trajectory of parameters and $\theta^0$ denotes the initial parameters. The gradient flow is a continuous-time limit of gradient descent.

Standard gradient descent can be viewed as a discrete-time approximation of the gradient flow 2 obtained through forward Euler discretization with step size $\eta^k$: $\theta^{k+1} = \theta^k - \eta^k \nabla_\theta f(\theta^k)$.

This continuous-time perspective provides valuable geometric intuition and theoretical tools for analyzing optimization dynamics, which we will leverage in developing our constrained optimization framework.

### 3.2 Constrained Optimization via Constraint-Aware Gradient Descent

We now consider the constrained problem

$$\min_\theta f(\theta) \quad \text{s.t.} \quad g(\theta) \le 0. \tag{3}$$

The unconstrained gradient flow 2 is insufficient here, since it may drive the trajectory into the infeasible region where $g(\theta) > 0$. A natural alternative is the projected gradient flow, which projects each update back onto the feasible set. However, the projection step is generally intractable for complex nonconvex constraints.

An alternative approach, studied in the literature from different perspectives, e.g., Muehlebach & Jordan (2021); Allibhoy & Cortés (2023), is to perturb the gradient flow with a corrective feedback term:

$$\frac{d\theta(t)}{dt} = -\nabla f(\theta(t)) + u(\theta(t)), \tag{4}$$

where $u(\theta(t))$ is chosen to minimally intervene so that the trajectory remains in the feasible set $\{\theta : g(\theta) \le 0\}$. Formally, the corrective term is obtained as the solution of the quadratic program

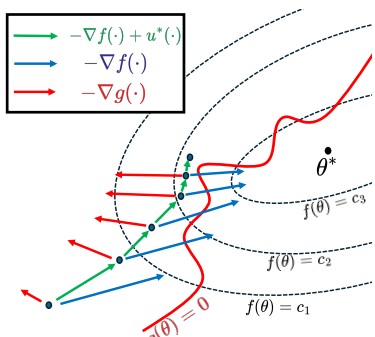

Figure 2: An illustrative example of constraint-aware gradient descent. When following the normal gradient descent direction $-\nabla f(\theta)$ would violate the linearized constraint, the correction term steers the updates to remain in the feasible set $g(\theta) \le 0$.

$$u(\theta(t)) = \{\arg\min_u \tfrac{1}{2}\|u\|^2 \text{ s.t. } \tfrac{d}{dt}g(\theta(t)) + \alpha g(\theta(t)) \le 0\}, \tag{5}$$

with design parameter $\alpha \ge 0$. This construction coincides with the well-established framework of *Control Barrier Functions* (CBFs) Ames et al. (2016), which guarantees forward invariance of the feasible set. In particular, if $u(\theta(t))$ is chosen according to 5, then any trajectory initialized strictly inside the feasible set, i.e., with $g(\theta(0)) < 0$, remains feasible for all $t > 0$. See Figure 2 for an illustrative example of the modified flow steps. The parameter $\alpha \ge 0$ controls how aggressively the barrier condition is enforced. Larger values of $\alpha$ allow trajectories to approach the boundary more quickly, while still preventing constraint violation. Equivalently, increasing $\alpha$ enlarges the feasible set of corrective controls in 5, which reduces the required intervention magnitude. In the limiting case where $\alpha \to \infty$, the barrier constraint becomes inactive, and the gradient projection flow method converges to the standard projected gradient flow dynamics Allibhoy & Cortés (2023).

The QP 5 admits a closed-form solution (see Section A.1 for derivation):

$$u^\star(\theta) = -\frac{\left(\alpha g(\theta) - \nabla_\theta f(\theta)^\top \nabla_\theta g(\theta)\right)_+}{\|\nabla_\theta g(\theta)\|^2} \nabla_\theta g(\theta)$$

Unlike projected gradient flow, which requires computing a potentially intractable projection onto the nonconvex feasible set and is only well defined for convex constraints, the correction term $u^\star(\theta)$ in Theorem 1 admits a simple closed-form expression that remains valid even when $g(\theta)$ is nonconvex.

**Constraint-Aware Gradient Descent**  Analogous to standard gradient descent as a discretization of the gradient flow  2, we discretize the modified flow  4 to obtain

$$\theta^{k+1} = \theta^k - \eta^k \left( \nabla_\theta f(\theta^k) + \frac{\left( \alpha^k g(\theta^k) - \nabla_\theta f(\theta^k)^\top \nabla_\theta g(\theta^k) \right)_+}{\|\nabla_\theta g(\theta^k)\|^2} \nabla_\theta g(\theta^k) \right)$$

where $\eta^k > 0$ is the step size and $(\cdot)_+$ returns the non-negative part of its argument. In the following theorem, we derive a sufficient condition for the step size $\eta^k$ that ensures anytime constraint satisfaction despite discretization errors.

**Theorem 3.1** (Constraint-Aware Gradient Descent). *Assume $f, g$ are continuously differentiable and that $\nabla g$ is $L_g$-Lipschitz, i.e.,*

$$\|\nabla g(\theta') - \nabla g(\theta)\| \leq L_g \|\theta' - \theta\| \quad \forall \, \theta, \theta'.$$

*Let $v(\theta) := -\nabla f(\theta) + u^\star(\theta)$, and consider the update $\theta^{k+1} = \theta^k + \eta^k v(\theta^k)$ with $\eta^k > 0$. Let*

$$\eta^{\max} := \frac{\alpha^k \, g(\theta^k) + \sqrt{(\alpha^k)^2 \, g(\theta^k)^2 - 2L_g \, g(\theta^k) \, \|v(\theta^k)\|^2}}{L_g \, \|v(\theta^k)\|^2}.$$

*If $g(\theta^k) < 0$ and $0 < \eta^k < \eta^{\max}$, then $g(\theta^{k+1}) < 0$. Consequently, if the condition holds for all $k$ and $g(\theta^0) < 0$, the iterates remain feasible at every step.*

*Proof.* As $\nabla g$ is $L_g$-Lipschitz, we can apply the standard descent lemma Beck (2017) on the iterates of $g$

$$g(\theta^{k+1}) \leq g(\theta^k) + \eta^k \nabla g(\theta^k)^\top v(\theta^k) + \frac{L_g \eta^{k^2}}{2} \left\| v(\theta^k) \right\|^2$$

$$\leq g(\theta^k) - \alpha^k \eta^k g(\theta^k) + \frac{L_g \eta^{k^2}}{2} \left\| v(\theta^k) \right\|^2,$$

where we have used the fact that $\nabla g(\theta^k)^\top v(\theta^k) \leq -\alpha g(\theta^k)$, based on the constraint of  5. As $g(\theta^k) < 0$, there exists an $\eta^{\max} > 0$ that zeros the right-hand side. This value is given by the positive root of the quadratic equation and is the value proposed by the theorem.

Consequently, by choosing $0 < \eta^k < \eta^{\max}$, given $g(\theta^k) < 0$, we guarantee $g(\theta^{k+1}) < 0$. It trivially follows that by satisfying this at every step and given an initial $g(\theta^0) < 0$, the constraint will always be satisfied. $\square$

Constraint-aware gradient descent integrates seamlessly with parameter-efficient fine-tuning methods such as Low Rank Adaptation (LoRA), in which only the calculation of the gradients $\nabla g$ and $\nabla f$ are modified and are natively handled by auto-differentiation frameworks. Moreover, the integration with memory optimization techniques such as gradient accumulation is also straightforward. That is, if a minibatch $B$ is broken down to $B_1, \cdots, B_n$, and gradients $\nabla g(\theta; B_1), \cdots, \nabla g(\theta; B_n)$ and $\nabla f(\theta; B_1), \cdots, \nabla f(\theta; B_n)$ are calculated in each step, respectively, then one simply needs to keep running summations $g = \frac{1}{n} \sum_{i=1}^n g(\theta; B_i), v_g = \frac{1}{n} \sum_{i=1}^n \nabla g(\theta; B_i)$, and $v_f = \frac{1}{n} \sum_{i=1}^n \nabla f(\theta; B_i)$. After every $B_i$ is iterated over, the update direction is simply

$$d = -v_f - \frac{(\alpha g - v_f^\top v_g)_+}{\|v_g\|^2} v_g$$

Furthermore, in specific instances of Problem  3 where the loss functions $f$ and $g$ operate on identical datasets ($\mathcal{D} = \mathcal{D}'$), our algorithm introduces negligible computational overhead. This efficiency stems from the computational structure of auto-differentiation frameworks, which leverage the chain rule for gradient computation. When the computation graphs of $f(\pi_\theta)$ and $g(\pi_\theta)$ differ only in their final operations applied to model outputs, the substantial majority of intermediate gradient calculations can be shared between the two functions, effectively amortizing the computational cost.

We present a high-level implementation of the stochastic variant of the constraint-aware gradient descent in Algorithm 1, formulated for the general case of distinct datasets $\mathcal{D}$ and $\mathcal{D}'$.

---

**Algorithm 1** Stochastic constraint-aware gradient descent for solving problem 3

---

1: **Input:** $\mathcal{D}, \mathcal{D}'$, batch sampling algorithm $\mathcal{R}$, reference parameters $\theta_{\text{ref}}$, loss functions $f$ and $g$, number of epochs $T$, barrier parameter $\alpha$, Optimizer $\mathcal{O}$
2: **Output:** Finetuned parameters $\theta$
3: **Initialize:** $\theta \leftarrow \theta_{\text{ref}}$
4: **for** $t = 1, \ldots, T$ **do**
5:     **for** $B_{\text{obj}}, B_{\text{cnt}}$ in $\mathcal{R}(\mathcal{D}, \mathcal{D}')$ **do**                          ▷ Get corresponding batch from each dataset.
6:         $\nabla f \leftarrow \nabla_\theta f(\pi_\theta, B_{\text{obj}}), \qquad \nabla g \leftarrow \nabla_\theta g(\pi_\theta, B_{\text{cnt}})$
7:         $d \leftarrow -\nabla f - \dfrac{\left(\alpha g(\pi_\theta, B_{\text{cnt}}) - \nabla f^\top \nabla g\right)_+}{||\nabla g||^2} \nabla g.$
8:         $\theta \leftarrow \mathcal{O}(d).$                          ▷ Use $d$ in optimizer as the descent direction.
9:     **end for**
10: **end for**
11: **Return:** $\theta$

---

## 4 CONSTRAINED OPTIMIZATION FOR LLM FINE-TUNING

Motivated by contemporary real-world deployment practices in commercial LLM services, we introduce a novel constrained fine-tuning optimization problem. Major commercial LLM providers, including OpenAI, Google's Vertex AI, and Cohere, offer proprietary models accessible through application programming interfaces (APIs) with fine-tuning capabilities that allow end-users to customize these models on custom proprietary datasets OpenAI; Cohere; Cloud. This functionality enables users to specialize general-purpose models for domain-specific tasks, improving performance on specialized applications while leveraging the substantial computational investments made by these providers towards their high-end models.

However, this capability introduces significant security and safety vulnerabilities. Malicious actors can exploit fine-tuning processes to induce harmful behaviors in models, extract sensitive information embedded within the base model, or circumvent existing safety mechanisms. Furthermore, even well-intentioned users may inadvertently introduce problematic content through noisy or insufficiently curated training data. Current industry practice for API-based fine-tuning typically involves post-hoc safety evaluations: upon completion of fine-tuning, providers apply safety checks to the resulting model. If the model fails to meet some safety specifications, the entire fine-tuned model is discarded and access is denied to the user.

This binary accept-or-reject approach is suboptimal for many reasons. First, it wastes computational resources when fine-tuning must be repeated with new data, and second, it provides a poor user experience when legitimate use cases are rejected due to incidental safety violations.

We propose a constraint-aware fine-tuning framework that addresses these limitations by incorporating safety considerations directly into the optimization process:

$$\min_\theta \quad \mathcal{L}_{\text{CE}}(\pi_\theta, \mathcal{D}) \qquad \text{subject to} \qquad \mathcal{L}_{\text{Safety}}(\pi_\theta, \mathcal{D}') \leq \varepsilon, \tag{6}$$

where $\mathcal{L}_{\text{CE}}(\pi_\theta, \mathcal{D})$ represents the standard cross-entropy fine-tuning loss on the user's desired dataset $\mathcal{D}$, and $\mathcal{L}_{\text{Safety}}(\pi_\theta, \mathcal{D}')$ quantifies safety violations over a representative evaluation dataset $\mathcal{D}'$ maintained by the provider.

In this work, we focus specifically on robustness against relearning attacks as a concrete instantiation of this framework. For this reason, we briefly discuss the problem of LLM unlearning and then motivate defenses against relearning attacks.

**LLM Unlearning** As LLMs are trained on web-scale corpora with, at times, minimal oversight on data quality, they may learn sensitive, copyrighted, or harmful information that must later be removed. The computational cost of retraining from scratch necessitates machine unlearning techniques that selectively remove knowledge without full retraining.

Unlearning is typically cast as a regularized optimization problem with loss functions $\mathcal{L}_{\text{fgt}}(\pi_\theta; \mathcal{D}_{\text{fgt}})$ and $\mathcal{L}_{\text{rtn}}(\pi_\theta; \mathcal{D}_{\text{rtn}})$ capturing forgetting quality and model utility, respectively,

$$\min_\theta \quad \mathcal{L}_{\text{fgt}}(\pi_\theta; \mathcal{D}_{\text{fgt}}) + \lambda \mathcal{L}_{\text{rtn}}(\pi_\theta; \mathcal{D}_{\text{rtn}}) \tag{7}$$

where $\lambda > 0$ is a trade-off parameter, $\mathcal{D}_{\text{fgt}}$ is a dataset containing samples to forget, and $\mathcal{D}_{\text{rtn}}$ is a dataset of normal samples that the model should perform well on even post-unlearning.

**Relearning Attacks.** Despite the relative success of unlearning algorithms in removing undesired knowledge from a model, several works have established that unlearned models are susceptible to relearning attacks Hu et al. (2024); Fan et al. (2025). Relearning attacks represent adversarial techniques that exploit residual traces of forgotten knowledge in unlearned models, demonstrating that successful unlearning procedures may remain vulnerable to targeted reconstruction efforts.

The fundamental premise of relearning attacks is that unlearning methods, while effective at suppressing access to specific knowledge during standard evaluation, may leave residual traces in the model's representational structure. The attack methodology is straightforward: adversaries perform standard fine-tuning on the unlearned model using samples from the original forget dataset $\mathcal{D}_{\text{fgt}}$. That is, by fine-tuning the unlearned model using a limited number of samples from $\mathcal{D}_{\text{fgt}}$, the model rapidly recovers its ability to generate the supposedly forgotten content, often achieving performance comparable to the pre-unlearning state.

For API providers operating unlearned models, these vulnerabilities pose serious risks. The originally unlearned knowledge was removed due to legitimate concerns. Successful relearning attacks compromise technical integrity, create potential legal liability, and undermine stakeholder confidence in responsible AI deployment. The threat model encompasses both malicious adversaries attempting to restore harmful capabilities and well-intentioned users who inadvertently trigger relearning through legitimate fine-tuning on similar data.

Our constrained fine-tuning framework provides a proactive defense by incorporating explicit constraints that monitor and limit the model's performance on forget-related content during fine-tuning, preventing relearning while enabling legitimate customization objectives. The relearning-safe fine-tuning procedure is thus cast as

$$\min_{\theta} \quad \mathcal{L}_{\text{CE}}(\pi_\theta, \mathcal{D}) \qquad \text{subject to} \qquad \mathcal{L}_{\text{fgt}}(\pi_\theta, \mathcal{D}'_{\text{fgt}}) \leq \varepsilon, \tag{8}$$

where $\mathcal{D}'_{\text{fgt}} \subseteq \mathcal{D}_{\text{fgt}}$, and $\varepsilon$ is an appropriate values based on $\mathcal{L}_{\text{fgt}}(\pi_{\theta^0}, \mathcal{D}_{\text{fgt}})$.

**Remark 4.1.** *The applications of our constrained fine-tuning formulation extend well beyond the relearning attack scenario, encompassing a broad range of safety-critical deployment contexts.*

*Chiefly, our algorithm easily handles unlearning through the recent formulation of unlearning as a constrained optimization by Entesari et al. (2025). We use this formulation in our experiments and see that our algorithm also conducts unlearning successfully. This is verified by comparing the unlearning metrics with those of Entesari et al. (2025).*

*Another particularly compelling application domain involves personalized AI assistants deployed on edge devices, where privacy-preserving personalization must be balanced against safety and regulatory compliance. As smaller language models become increasingly capable and edge deployment becomes more prevalent, users expect AI systems that adapt to their specific needs while maintaining appropriate behavioral boundaries.*

*Our framework addresses this challenge by treating safety requirements as constraints rather than post-hoc filters. This approach enables gradient-based optimization that naturally balances personalization objectives against safety requirements, potentially achieving superior outcomes compared to traditional binary accept-or-reject paradigms that provide no guidance for remediation when safety checks fail.*

*We defer comprehensive exploration of these extended applications and their specific technical requirements to future work, focusing here on establishing the foundational methodology and demonstrating its effectiveness in the relearning attack mitigation scenario.*

## 5 EXPERIMENTS

**Setup** We focus on relearning attacks on the Task of Fictitious Unlearning (TOFU) Maini et al. (2024). That is, we take a model that has undergone unlearning on the retain90/forget10 subset of the TOFU dataset, and then study finetuning attempts using the same forget dataset. To study the different effects of our framework on finetuning, we consider three finetuning setups:

- Pure adversarial: The finetuning data is the forget subset from the unlearning task, i.e., $\mathcal{D} = \mathcal{D}_{\text{fgt}}$.

- Normal data: The finetuning data non-adversarial with no element from $\mathcal{D}_{\text{fgt}}$, i.e., $\mathcal{D} = \mathcal{D}_{\text{nor}}$.

- Adversarial shuffle: The finetuning data is made up of both non-adversarial data and also adversarial forget data, i.e., $\mathcal{D} = \mathcal{D}_{\text{fgt}} \cup \mathcal{D}_{\text{nor}}$.

For $\mathcal{D}_{\text{fgt}}$, we utilize the full `forget10` data subset from TOFU. As for the normal non-adversarial data $\mathcal{D}_{\text{nor}}$, there are many publicly available datasets that could be utilized. However, each dataset could potentially introduce mismatches from data format to domain knowledge. As such, in this work, we propose to use a cloned version of the TOFU benchmark to prevent potential issues. That is, using the same guidelines outlined in Maini et al. (2024), we prompt OpenAI's ChatGPT 4.1 model to create a dataset of fictitious authors with a set of questions and answers on the authors. By using a dataset in the same Question-Answer format and domain as that of the unlearned data, we reduce the potential for uncontrolled factors interfering with our analysis. Slightly different from Maini et al. (2024), we ask the LLM to create 25 questions per fictitious author (instead of the normal 20) and retain 5 questions from each author for the evaluation subset.

As per unlearning literature precedent, we utilize the Llama family of models and use Llama 3.2 1B and 3B and Llama 3.1 8B models for our experiments. Due to computational limitations, we utilized parameter-efficient fine-tuning techniques for our experiments and used Low-Rank Adaptation (LoRA) for all models. The details of the finetuning can be found in the Appendix.

We compare the success of our proposed method against the sole baseline of no defense, i.e., normal fine-tuning, in which no defense mechanism is applied and a single cross-entropy loss on the model's completions is calculated.

To acquire models that have undergone unlearning, we utilize our Constraint-Aware Fine-tuning methodology. For this, we utilize the recent work of Entesari et al. (2025), which formulates machine unlearning as a constrained optimization problem. Whereas Entesari et al. (2025) uses a primal-dual framework to solve the unlearning problem, we propose to use our algorithm on it. We see in Table 1 that our methodology also performs exceptionally well in unlearning (Constraint-Aware Unlearning), but we leave rigorous comparisons on this matter to future work.

**Evaluation** To evaluate our framework, we measure several factors. First, to study whether fine-tuning has been successful, we calculate the ROUGE-L recall score on the train and validation subsets of $\mathcal{D}_{\text{nor}}$. Successful fine-tuning should yield models with high ROUGE scores. Second, to measure the success or failure of relearning attacks and the defense against them, we utilize the forgetting metrics ROUGE-L and Probability from the TOFU evaluation suite for the forget subset. Models that have undergone unlearning would have low values for these metrics, and a high ROUGE metric could be a sign of knowledge retention from the forget subset $\mathcal{D}_{\text{fgt}}$. We aggregate these two metrics into a single *forgetting* metric, which is calculated as their harmonic mean. Third, to study the effect of finetuning on prior tasks, we report the *model utility* metric from the TOFU evaluation suite. This metric represents the model performance on the TOFU retain90 task and other notions of utility as described in Maini et al. (2024).

**Results** We present our main results in Table 1 for all three models and three fine-tuning scenarios. Studying rows corresponding to the normal fine-tuning setup, i.e., $\mathcal{D} = \mathcal{D}_{\text{nor}}$, reveals that across all models, our algorithm does not negatively impact the normal fine-tuning procedure and the model learns the desired information. This is evident by comparing the ROUGE scores across the naive and constraint-aware fine-tuning rows and contrasting them with the starting *unlearned* model. A very high ROUGE-L Train score denotes a form of memorization, which happens with both fine-tuning algorithms. It is expected that ROUGE-L Val should be smaller, and we see very similar numbers using either fine-tuning algorithm.

Next, the adversarial shuffle entries, i.e., $\mathcal{D} = \mathcal{D}_{\text{NF}}$, reveal the strength of our methodology. Comparing the ROUGE scores reveals that both fine-tuning algorithms learn the normal data pattern and have similar performance on the ROUGE-L Val metric. However, studying the *forgetting* score reveals that the naive fine-tuning algorithm has done so whilst regaining its knowledge on the forget dataset, whereas our algorithm has successfully defended against this attack and has maintained the same performance as that of the starting *unlearned* model. We see this similar pattern in the case of the pure adversarial attack, i.e., $\mathcal{D} = \mathcal{D}_{\text{fgt}}$.

Table 1: Relearning attacks on Llama models unlearned on the TOFU dataset (`retain90/forget10`). All metrics are bounded in $[0, 1]$ and the arrows indicate whether larger numbers are better or not. The second column denotes the dataset used for the fine-tuning task. When $\mathcal{D} = \{\}$, no fine-tuning is done and these entries establish baselines on the starting models. 'target' is the model that has not been unlearned, with knowledge of $\mathcal{D}_{\text{rtn}}$ and $\mathcal{D}_{\text{fgt}}$, 'retrained' only has knowledge of $\mathcal{D}_{\text{rtn}}$, and 'unlearned' is the target model that has undergone unlearning. We define $\mathcal{D}_{\text{NF}} = \mathcal{D}_{\text{nor}} \cup \mathcal{D}_{\text{fgt}}$. For each experimental setup, i.e., each 'naive-Ours' pair, if one method significantly outperforms the other, we bolden that entry.

| | $\mathcal{D}$ | Methodology | Constraint-Aware Unlearning (Ours) | | | |
| | | | ROUGE-L Train↑ | ROUGE-L Val↑ | **Forgetting** ↓ | Model Utility ↑ |
|---|---|---|---|---|---|---|
| Llama 3.2 1B | $\{\}$ | target | 0.5673 | 0.5747 | 0.8492 | 0.5992 |
| | | retrained | 0.5727 | 0.5739 | 0.1778 | 0.5911 |
| | | unlearned | 0.5159 | 0.5626 | 0.0000 | 0.5742 |
| | $\mathcal{D}_{\text{nor}}$ | naive | 0.9198 | 0.6978 | 0.0000 | 0.4430 |
| | | Ours | 0.9049 | 0.7002 | 0.0000 | 0.4427 |
| | $\mathcal{D}_{\text{NF}}$ | naive | **0.8905** | 0.7001 | 0.8371 | 0.4947 |
| | | Ours | 0.7617 | 0.6924 | **0.0003** | 0.4561 |
| | $\mathcal{D}_{\text{fgt}}$ | naive | 0.5838 | 0.5943 | 0.6842 | 0.5412 |
| | | Ours | 0.5477 | 0.5808 | **0.0001** | 0.5652 |
| Llama 3.2 3B | $\{\}$ | target | 0.5797 | 0.5870 | 0.9384 | 0.6661 |
| | | retrained | 0.5547 | 0.5890 | 0.1878 | 0.6498 |
| | | unlearned | 0.5440 | 0.5578 | 0.0000 | 0.6253 |
| | $\mathcal{D}_{\text{nor}}$ | naive | 0.9100 | 0.7087 | 0.0045 | 0.4768 |
| | | Ours | 0.9381 | 0.7081 | 0.0000 | 0.4756 |
| | $\mathcal{D}_{\text{NF}}$ | naive | **0.9506** | 0.7070 | 0.9086 | 0.5607 |
| | | Ours | 0.7730 | 0.6935 | **0.0004** | 0.5145 |
| | $\mathcal{D}_{\text{fgt}}$ | naive | 0.6156 | 0.5927 | 0.7548 | 0.6001 |
| | | Ours | 0.6026 | 0.5823 | **0.0001** | 0.6259 |
| Llama 3.1 8B | $\{\}$ | target | 0.6214 | 0.6004 | 0.9911 | 0.6276 |
| | | retrained | 0.5918 | 0.6072 | 0.1665 | 0.6461 |
| | | unlearned | 0.5879 | 0.5984 | 0.0000 | 0.7068 |
| | $\mathcal{D}_{\text{nor}}$ | naive | **0.9491** | 0.7149 | 0.4901 | 0.4977 |
| | | Ours | 0.8764 | 0.7255 | **0.0000** | 0.2423 |
| | $\mathcal{D}_{\text{NF}}$ | naive | **0.9665** | 0.7123 | 0.9858 | **0.5841** |
| | | Ours | 0.7711 | 0.7230 | **0.0003** | 0.3089 |
| | $\mathcal{D}_{\text{fgt}}$ | naive | **0.6050** | **0.5936** | 0.9976 | **0.6059** |
| | | Ours | 0.0000 | 0.0005 | **0.0000** | 0.0000 |

We see that for the larger 8B model, the algorithm successfully defends against the attacks as well. In the pure adversarial setup $\mathcal{D}_{\text{fgt}}$, our algorithm has successfully defended against the relearning attack, but as there was no real fine-tuning objective at hand and since the capacity of the model is much greater, our fine-tuning algorithm has also affected the utility of the model. If this is undesired, it can be avoided with further hyperparameter modifications.

## 6 CONCLUSION

We introduced a security-constrained fine-tuning framework that formulates safety requirements as explicit constraints rather than heuristics or post-hoc checks. Our constraint-aware gradient descent algorithm efficiently handles such constraints through a modified flow application, yielding closed-form solutions at each optimization step. Experiments on defending against relearning attacks on the TOFU dataset using a range of Llama models demonstrate robust defense against relearning attacks while maintaining legitimate fine-tuning performance.

While we focused on relearning defense, our framework applies broadly to safety-constrained model customization. Future work includes comprehensive exploration of extended applications such as personalized AI assistants on edge devices, rigorous comparison with existing unlearning baselines, and evaluation across diverse safety-critical deployment scenarios. We defer these extensions to establish the foundational methodology demonstrated here.

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

# A    APPENDIX

## A.1    DERIVATION OF CLOSED FORM SOLUTION OF PROBLEM 5

*Proof.* As problem 5 is convex in $u$ and has only a single affine constraint, strong duality holds Boyd & Vandenberghe (2004). As such, the solution can be acquired through the KKT conditions. The Lagrangian for this problem is

$$\mathcal{L}(u, \lambda) = \frac{1}{2}\|u(\theta(t))\|_2^2 + \lambda\left(\alpha g(\theta) + \nabla_\theta g(\theta)^\top u(\theta) - \nabla_\theta g(\theta)^\top \nabla_\theta f(\theta)\right).$$

Solving for $\nabla_u \mathcal{L}(u, \lambda) = 0$ yields $u(\theta) = -\lambda \nabla_\theta g(\theta)$. Using this in the primal feasibility condition yields that

$$\lambda \geq \frac{\alpha g(\theta) - \nabla_\theta f(\theta)^\top \nabla_\theta g(\theta)}{\|\nabla g(\theta)\|^2}.$$

Moreover, we know that $\lambda \geq 0$. Putting all these together, along with the complementary slackness condition, we find that

$$\lambda^\star = \frac{\left(\alpha g(\theta) - \nabla_\theta f(\theta)^\top \nabla_\theta g(\theta)\right)_+}{\|\nabla_\theta g(\theta)\|^2}.$$

This concludes the proof. □

## A.2    EXPERIMENTS

### A.2.1    IMPLEMENTATION DETAILS

Based on the memory requirements of different experiments, we conduct our experiments in two setups. For the experiments using the LLAMA 3.2 1B/3B models, we use a single A100 GPU with 40GB of memory. For the LLAMA 3.1 8B model, we use a single H100 80 GB GPU.

We build our code on top of the `open-unlearning` Dorna et al. (2025) GitHub repository[1]. The repository provides target and retrained models for the TOFU dataset. To perform unlearning, we utilize mostly the default parameters of the repository with the addition of LoRA. We use LoRA with $\alpha = r = 128$ for unlearning. We apply the LoRA using the HuggingFace library to all linear layers. For the 1B and 3B models, we use a learning rate of $0.001$ (with the default warm-up and decay), but for the 8B model we use a learning rate of $0.00001$. We use a fixed effective batch size of 32. For the constrained unlearning framework, we set $\varepsilon = 0.3$ on the retention loss, as per Entesari et al. (2025).

For all constraint-aware gradient descent implementations, we use $\alpha^k = 1/\eta^k$, i.e., at each step, $\alpha$ is the reciprocal of the current learning rate. This choice is the maximal value allowed under one-step approximation of the time derivative in problem 5, i.e., $\frac{g(\theta^{k+1}) - g(\theta^k)}{\eta^k} + \alpha g(\theta^k) \leq 0$. For $g(\theta^k) \leq 0$, in order to have $g(\theta^{k+1}) \leq 0$, we must have $\alpha \leq \frac{1}{\eta^k}$.

For the fine-tuning experiments, we utilize LoRA with $\alpha = r = 64$ applied on all linear layers. For the 1B and 3B models, we use a learning rate of $0.0001$, and a learning rate of $0.00005$ for the 8B model. Based on the value of the forget loss for the unlearned models $\mathcal{L}_{\text{fgt}}(\pi_{\theta^0}, \mathcal{D}_{\text{fgt}})$, we use a fixed $\varepsilon = 10$ for the constraint of the defense against relearning.

### A.2.2    ABLATION

An important question with practical implications for the implementation of safety constraints in LLM applications is the effect of the size of the dataset $\mathcal{D}'$ needed for the constraint. In our setup for the experiments in Table 1, we used the full forget10 subset from the TOFU dataset for the constraint, and at each iteration, a minibatch would be sampled from this dataset and used for our algorithm (as per Algorithm 1). In this section, we study the effect of the available samples in $\mathcal{D}'$ on the performance of our algorithm.

---

[1]https://github.com/locuslab/open-unlearning

Table 2: Effect of the size of the auxiliary dataset used for the constraint. All metrics are bounded in $[0, 1]$ and the arrows indicate whether larger numbers are better or not. The second column denotes the dataset used for the fine-tuning task. $|\mathcal{D}'|/|\mathcal{D}_{\text{fgt}}|$ represents the ratio of the size of the auxiliary dataset vs the full size. A value of $|\mathcal{D}'|/|\mathcal{D}_{\text{fgt}}| = 0$ represents the naive fine-tuning algorithm and $|\mathcal{D}'|/|\mathcal{D}_{\text{fgt}}| = 1$ represents our full algorithm, as used in table Table 1. We let $\mathcal{D}_{\text{NF}} = \mathcal{D}_{\text{nor}} \cup \mathcal{D}_{\text{fgt}}$

| | | | Constraint-Aware Unlearning (Ours) | | | |
|---|---|---|---|---|---|---|
| | $\mathcal{D}$ | $\lvert\mathcal{D}'\rvert/\lvert\mathcal{D}_{\text{fgt}}\rvert$ | ROUGE-L Train↑ | ROUGE-L Val↑ | Forgetting ↓ | Model Utility ↑ |
| Llama 3.2 1B | $\mathcal{D}_{\text{nor}}$ | 0 | 0.9198 | 0.6978 | 0.0000 | 0.4430 |
| | | 0.2 | 0.9034 | 0.6980 | 0.0000 | 0.4418 |
| | | 0.5 | 0.9398 | 0.6992 | 0.0000 | 0.4468 |
| | | 1 | 0.9049 | 0.7002 | 0.0000 | 0.4427 |
| | $\mathcal{D}_{\text{NF}}$ | 0 | 0.8905 | 0.7001 | 0.8371 | 0.4947 |
| | | 0.2 | 0.7763 | 0.6917 | 0.4128 | 0.4624 |
| | | 0.5 | 0.7559 | 0.6868 | 0.0047 | 0.4597 |
| | | 1 | 0.7617 | 0.6924 | 0.0003 | 0.4561 |
| | $\mathcal{D}_{\text{fgt}}$ | 0 | 0.5838 | 0.5943 | 0.6842 | 0.5412 |
| | | 0.2 | 0.5491 | 0.5800 | 0.0007 | 0.5654 |
| | | 0.5 | 0.5753 | 0.5707 | 0.0001 | 0.5688 |
| | | 1 | 0.5477 | 0.5808 | 0.0001 | 0.5652 |
| Llama 3.2 3B | $\mathcal{D}_{\text{nor}}$ | 0 | 0.9100 | 0.7087 | 0.0045 | 0.4768 |
| | | 0.2 | 0.9246 | 0.7058 | 0.0000 | 0.4753 |
| | | 0.5 | 0.9464 | 0.7086 | 0.0000 | 0.4787 |
| | | 1 | 0.9381 | 0.7081 | 0.0000 | 0.4756 |
| | $\mathcal{D}_{\text{NF}}$ | 0 | 0.9506 | 0.7070 | 0.9086 | 0.5607 |
| | | 0.2 | 0.7933 | 0.7016 | 0.5816 | 0.5320 |
| | | 0.5 | 0.8350 | 0.6983 | 0.0349 | 0.5104 |
| | | 1 | 0.7730 | 0.6935 | 0.0004 | 0.5145 |
| | $\mathcal{D}_{\text{fgt}}$ | 0 | 0.6156 | 0.5927 | 0.7548 | 0.6001 |
| | | 0.2 | 0.5588 | 0.5819 | 0.0019 | 0.6269 |
| | | 0.5 | 0.5745 | 0.5863 | 0.0003 | 0.6225 |
| | | 1 | 0.6026 | 0.5823 | 0.0001 | 0.6259 |

The TOFU dataset has 20 questions per fictitious author. We study the effect of using only 10 and 4 questions per author in Table 2, denoted with $|\mathcal{D}'|/|\mathcal{D}_{\text{fgt}}|$ equal to 0.5 and 0.2, respectively. We repeat the rows corresponding to the 'naive' and 'Ours' entries from Table 1 in rows with $|\mathcal{D}'|/|\mathcal{D}_{\text{fgt}}|$ equal to 0 and 1, respectively, for ease of comparison.

As expected, the larger $\mathcal{D}'$ is, the better. We see that for this experimental setup, a ratio of 50% is enough, and our algorithm prevents relearning by having access to half of the questions from each author. Making the dataset further smaller hurts the algorithm's capacity to keep the forgetting score low, and with only 4 questions per author, there is some relearning done by the model.

