# OpenReview forum: "Security-Constrained Fine-tuning: Preventing Knowledge Restoration in Unlearned Models"
_ICLR.cc/2026/Conference — Submitted to ICLR 2026_

### Official Review · Reviewer_ad2Q · 2025-10-22

**Soundness:** 2
**Presentation:** 3
**Contribution:** 2
**Rating:** 2
**Confidence:** 4

**Summary:**

This paper tackles the problem of LLMs relearning forgotten knowledge after machine unlearning. The authors propose a constrained fine-tuning framework that treats safety/unlearning as a hard constraint during optimization rather than a post-hoc check. They derive a constraint-aware gradient update based on safe gradient flow, preventing the model from reducing the “forget loss” during fine-tuning, thereby blocking relearning attacks. Experiments on LLaMA models under the TOFU benchmark show that the method effectively preserves unlearning while maintaining utility on normal data.

**Strengths:**

1. The paper formulates the problem as a constrained optimization task and introduces an optimization algorithm to address it, providing a meaningful and clear perspective.
2. Theoretical guarantees are thoroughly presented, with detailed proofs ensuring that the constraint is satisfied during optimization.
3. The problem is modeled in a clear manner, and the algorithm is described with mathematical rigor and precision.

**Weaknesses:**

1. The constrained optimization formulation and gradient correction mechanism largely overlap with existing work [1]. Thus, the core optimization idea is not conceptually new.
2. The problem positioning is somewhat unclear. Although the paper frames the method as safety- or unlearning-related, the focus is specifically on defending against relearning after unlearning. This is neither a purely unlearning problem nor a standard safety problem, and the connection to safety is indirect. Moreover, for the relearning attack setting, no baseline methods are provided, making it difficult to assess the actual effectiveness of the approach.
3. The experiments are limited in scope. The paper only evaluates relearning mitigation, without including complexity analysis, runtime evaluation, ablation studies, or sensitivity to hyperparameters. As a result, the empirical validation is insufficient.

[1] Allibhoy, A., & Cortés, J. (2023). Control-barrier-function-based design of gradient flows for constrained nonlinear programming. IEEE Transactions on Automatic Control, 69(6), 3499-3514.

**Questions:**

1. Since the paper frames the method in the context of safety-constrained fine-tuning, could the authors provide results on safety-centric benchmarks or tasks (e.g., harmful content prevention, jailbreak robustness, refusal alignment), rather than only focusing on the relearning scenario after unlearning? This would help clarify whether the method contributes to safety itself, rather than only mitigating relearning.
2. The optimization framework appears strongly related to prior work on constrained gradient flows and control barrier functions. Are the contributions mainly in applying this framework to the relearning/unlearning setup, or does the method introduce any fundamentally new theoretical ideas, constraint formulations, or algorithmic mechanisms beyond existing constrained optimization literature?

---

> ### Author Response · Authors · 2025-12-01
>
> Thank you for your review and for recognizing the **clarity** of our problem formulation and the **rigor** of our constrained optimization framework. We appreciate your acknowledgment of the theoretical guarantees and the **meaningful perspective our approach** provides for preventing relearning attacks. In this rebuttal, we address your concerns and provide additional clarification.
>
> Regarding the similarity to [1], we explicitly acknowledge this connection in our manuscript. Our work provides two main extensions: First, we establish the framework of [1], which was developed for gradient flow and continuous-time systems, for the discrete dynamics inherent to practical optimization algorithms. Second, we introduce a novel class of problems for LLMs under the framework of security-constrained fine-tuning and demonstrate how our technique effectively addresses such constrained optimization problems.
>
> Regarding experimental coverage, please refer to our global comments where we have added two new experimental setups (LLM unlearning and aligned fine-tuning) and an additional regularization baseline for the relearning defense task.

---

### Official Review · Reviewer_VdF2 · 2025-10-30

**Soundness:** 2
**Presentation:** 2
**Contribution:** 2
**Rating:** 2
**Confidence:** 4

**Summary:**

This paper aims to address the retraining attack towards machine unlearning on large language models (LLMs). The background is that LLM maintainers may need to unlearn some training samples for a given LLM. However, as revealed by existing works, such unlearning is incomplete and there remain some residuals in LLMs; the unlearned samples can be rapidly recovered by simple fine-tuning. This procedure is named the retraining attack. To address this problem, they propose a new constrained-based optimization method. The core idea is to formalize the constraint terms beyond the original objective function, and such a subjection condition can be derived to an extra gradient optimization direction. Theoretically, the authors prove that under specific conditions their method can ensure the satisfaction of the condition under the promise of control barrier function. They provide some experiments on LLM unlearning using the TOFU dataset for empirical validation.

**Strengths:**

+ This paper considers an interesting problem, i.e., how to achieve better unlearning with higher quality.
+ The proposed algorithm is somewhat novel and comes with a theoretical guarantee.

**Weaknesses:**

+ The organization of this paper seems terrible. There are lots of unnecessary repetitions between Section 4 and Sections 1 & 2. I strongly suggest the authors consider reformatting Section 4 by removing all unnecessary emphasis on contributions (there are too many), related work introductions (you have already clarified them before), and Remark 4.1.
+ The threat model explored in this paper is unclear. When I consider it, I think it could be impractical. Retraining attack towards unlearning, in my opinion, is more about unlearning analysis than a realistic attack. Under the retraining attack scenario, who is the adversary? Is it the model maintainer or user? Who is the defender? Does this paper aim to provide a more robust/powerful unlearning than existing methods, making the forgotten dataset harder to recover, or is this strategy added exactly at the fine-tuning (i.e., re-training attack) process? If it is the latter, why would the attacker choose to use your algorithm (which is a defense) in his/her retraining attack procedure?
+ Significant details are missing. Even after reading the whole paper, I cannot find the exact formulation of $L_{fgt}$ shown in Equation (8). You can skip the definition of $L_{ce}$ as it is at least noted as $CE$, letting me know that it is cross-entropy-based fine-tuning, but what are the formulations of $L_{fgt}$, $L_{rtn}$, and $L_{Safety}$ used in your experiments?
+ Regarding experiments, there are some concerns too. The first one is that the baseline is overly simple. The experiments only compare with one baseline, i.e., no defense. I suggest the authors include existing unlearning methods that consider the residual (if there are any) and some simple baselines. For instance, they could use a simple convex combination of the objective function and the subjected objective function as the training loss as a baseline.
+ My second concern about the experiments is their results. Under the same model backbone with different datasets, why is {} significantly lower than others? From 57 to 90, it is a really big gap and not reasonable. Also, for {}, why does retrained exhibit a higher forgetting score than unlearned? I regard the retrained line as the results of a retrained model, which should be an ideal situation, i.e., the upper bound of any unlearning methods. Additionally, why does llama3.1-8B on $D_{nor}$ exhibit a forgetting score of 49 under naive fine-tuning? For the same setting with other models, it is zero.
+ Minor issues. There are many misuses of `citep` and `citet` in the paper. Also, I cannot get timely explanations of notations. For instance, what is $()_+$ shown in line 210? I cannot get the explanation until I read the next equation which also uses this notation.

**Questions:**

Please refer to the weakness.
If there are some significant misunderstandings, I'd like to raise my score.

---

> ### Author Response · Authors · 2025-12-01
>
> Thank you for your review and for acknowledging the importance of addressing relearning attacks in LLM unlearning. We appreciate your recognition of the **novelty of our constrained optimization approach** and the theoretical guarantees we provide. In this rebuttal, we address your concerns and provide additional clarification.
>
> For ease of cross-referencing, we number the weaknesses and reply to each accordingly.
>
> 1. We appreciate this feedback. We recognize that our emphasis on the framework's applicability became redundant in places. The revised manuscript addresses this issue by removing unnecessary remarks and streamlining the presentation.
>
> 2. We apologize for any ambiguity in our threat model description. We have clarified this in our global comments and provide additional elaboration here:
>
>     An LLM provider (such as OpenAI) offers API access allowing users to fine-tune proprietary models $\pi$ on arbitrary datasets $D$. After fine-tuning, the provider grants API access to the fine-tuned model $\pi_\mathrm{ft}$. The LLM provider acts as the defender and employs our proposed security-constrained fine-tuning framework. The provider's objective is to prevent $\pi_\mathrm{ft}$ from becoming harmful due to malicious samples in $D$.
>
> 3. We apologize for the lack of clarity regarding these loss functions. In the main text, we intentionally left $L_\mathrm{safety}$ and related terms general, as they can be instantiated with any appropriate loss function depending on the application. For instance, $L_\mathrm{safety}$ could be a reward model measuring harmlessness, helpfulness, or other safety properties.
>
>     For our specific experiments, as noted in Remark 4.1, we adopt the framework from Entesari et al. (2025) "Constrained Entropic Unlearning: A Primal-Dual Framework for Large Language Models." In the relearning attack defense setting, $L_\mathrm{safety}$ (chosen as $L_\mathrm{fgt}$) corresponds to the logit-flattening loss proposed in that work.
>
> 4. We thank the reviewer for their suggestion. We have added the suggested baseline and have provided results in the global comment.
>
> 5. Unfortunately, it seems that due to markdown rendering issues in the original review, we cannot fully parse this comment. However, we address what we believe to be the main concern.
>
>     The 'forgetting' score is a metric that attempts to measure unlearning through proxy measures such as loss function values. Since unlearned models are trained using such loss functions, they can sometimes achieve better scores than retrained models that have not been subject to specific loss-based fine-tuning. This reflects a fundamental challenge in the field: measuring true forgetting is difficult, and the literature relies on surrogate metrics that approximate the desired property.
>
>     Regarding the higher forgetting score on $D_\mathrm{nor}$ when using the 8B model, we offer the following explanation. The 8B model exhibits different behavior across several metrics, which may stem from its higher capacity, different unlearned starting point, and associated stochasticity. More importantly, we emphasize the surrogate nature of the forgetting metric and the similarity between $D_\mathrm{nor}$ and the forget dataset $D_\mathrm{for}$—both are QA datasets about authors. When the model is fine-tuned on $D_\mathrm{nor}$, it generally improves at answering questions, including those in $D_\mathrm{for}$, even though the answers may be incorrect. This leads to higher ROUGE scores, which contribute to the forgetting metric calculation.
>
> 6. We thank the reviewer for pointing out these issues. We will fix them in the revised manuscript and ensure all notation is properly defined.

---

### Official Review · Reviewer_FKwD · 2025-11-01

**Soundness:** 3
**Presentation:** 3
**Contribution:** 2
**Rating:** 4
**Confidence:** 4

**Summary:**

The paper proposes a novel framework for security-constrained fine-tuning of large language models (LLMs), motivated by the vulnerability of unlearned models to “relearning attacks,” where adversaries reintroduce deliberately forgotten knowledge via fine-tuning. The authors formulate fine-tuning as a constrained optimization problem, where standard objectives are optimized subject to constraints that explicitly prevent restoration of unwanted information. They introduce a Constraint-Aware Gradient Descent algorithm, which efficiently approximates intractable nonconvex constraints by leveraging first-order Taylor expansions to yield closed-form update steps, and demonstrate empirically—using the TOFU benchmark and Llama models—that their approach can block relearning attacks while largely maintaining legitimate task performance.

**Strengths:**

- Principled Approach: The re-casting of safe fine-tuning as an explicit constrained optimization problem is well-motivated and offers a methodological improvement over heuristic or purely post-hoc safety checks. By directly incorporating security or compliance constraints into the training objective, the approach has clear benefits for both practical deployment (e.g., commercial LLM APIs) and theoretical understanding.

- Algorithmic Clarity and Efficiency: The closed-form corrective step for constraint satisfaction is derived rigorously (as presented in Section 3.2 and detailed in the Appendix), offering theoretical guarantees (Theorem 3.1) under reasonable assumptions and practical implementation advantages—especially when constraints and objectives share structure (see discussion following Eq. 5 and Algorithm 1).

- Strong Empirical Demonstration: The experimental evaluation (Section 5 and Table 1) convincingly shows that, on the TOFU benchmark, the proposed method can block relearning attacks nearly completely (Forgetting metric remains $\sim0$ for Ours vs. $\gg0$ for naive methods), while still permitting high performance on legitimate customization objectives (ROUGE-L and Model Utility metrics remain competitive in non-adversarial scenarios). The ablation in Table 2 further details the practical sensitivity to constraint dataset size.

- Good Use of Visualizations: Figure 1 concisely and effectively illustrates the threat model, progression from unlearning to relearning attacks, and the intended protection offered by the proposed method—contributing to clarity of exposition for readers less familiar with the problem space.

- Methodological Rigor: Mathematical aspects are well worked out, with the key corrective update and step size derived and justified; explicit conditions for anytime constraint satisfaction in practice (Theorem 3.1, Eq. 8) are provided and explained.

- Reproducibility and Transparency: Implementation details (Appendix A.2.1) are given with explicit hyperparameters, LoRA setup, hardware environment, and use of the open-unlearning codebase, supporting practical reproducibility.

**Weaknesses:**

- Limited Scope of Experiments and Baselines: The experimental analysis is limited primarily to the TOFU benchmark and Llama models across three sizes, with a single dominant baseline (naive fine-tuning, i.e., no defense). The exclusion of additional recent competitive defenses, especially those specifically designed for robust unlearning (e.g., MUSE, OpenUnlearning, and methods from Fan et al. 2025; Zhan et al. 2023), seriously weakens claims of broad practical superiority or generalizability.

- Missing Direct Comparison to Recent Benchmarks: Notable recent benchmarks for knowledge persistence and unlearning, such as RWKU (Jin et al., 2024), are not employed, and related-state-of-the-art methods uncovered in the provided search context (e.g., Are We Really Unlearning? by Hsu et al., 2025) are absent from both related work discussion and experimental evaluation. This omission leaves it unclear whether the framework’s strengths generalize beyond TOFU and standard Llama models.

- Mathematical/Algorithmic Questions: While the formulation is clear, Section 3.2 and the proof in Appendix A.1 depend on $g$ and $f$ being continuously differentiable and on the $L_g$-Lipschitz property holding. In the LLM setting, with high-dimensional, non-smooth parameter landscapes and potential batch-level stochasticity, how these assumptions translate and whether the guarantees persist are not empirically probed. Moreover, constraint satisfaction hinges on choices of $\alpha^k$ and $\eta^k$ (see Eq. 8 and Appendix A.2.1), but little practical guidance beyond using $\alpha^k=1/\eta^k$ is given—leaving questions about robustness to hyperparameter selection.

- Underdeveloped Discussion of Trade-offs: Table 1 shows that, especially for the largest (8B) Llama model and in pure adversarial settings, the constraint-aware method can result in degraded Model Utility—even a collapse to zero. The explanation is brief: “this can be avoided with further hyperparameter modifications”—but no details or concrete mitigation strategies are provided. This trade-off between safety and utility deserves much deeper quantitative and qualitative investigation. Furthermore, Figure 1, while helpful as a conceptual overview, could benefit from added detail or supplementary visualizations (such as training dynamics or constraint violation plots).

- Insufficient Exploration of Broader Applicability: While the paper claims broad generalizability beyond the relearning scenario (see Remark 4.1), the methodology is only demonstrated concretely for the relearning/TOFU setting. No empirical evidence or pilot results are presented for privacy-preserving personalization, regulatory constraint scenarios, or alignment-preserving fine-tuning—limiting the demonstrable scope of utility at this stage.

- Incomplete Literature Positioning: Although the related work section (Section 2) is detailed, several recent empirical and benchmark efforts (RWKU, Hsu et al. 2025) and especially works exploring the persistence of residual knowledge in unlearning (Wei et al. 2023, Jin et al. 2024) are not cited or contrasted—even as these have material implications for evaluation and baseline comparison.

- Algorithmic Clarity and Pseudocode: While Algorithm 1 provides a reasonable high-level - outline, some notational choices (e.g., the usage of inner products and matrix norms in update directions) and practical training adjustments are only briefly described. More detailed pseudo-code or reference implementations would further improve clarity and usability.

- Constraint Dataset Selection and ‘Coverage’: Table 2 ablates constraint dataset size but relies on the TOFU ‘forget10’ subset. There is no analysis of the effect of distribution mismatch between constraint/evaluation and potential adversarial data, or guidelines for constructing constraint sets in more realistic, open-set environments.

- Potential for Over-constraining: With a strong constraint dataset or high $\alpha$, the method may lead to a substantial loss of model utility for non-adversarial objectives (as partially seen in Table 1 for the 8B model). The balance between utility and safety is acknowledged as tunable but not robustly quantified.

**Questions:**

- Generality to Non-TOFU Datasets & Real-World Scenarios: Beyond TOFU and synthetic ‘cloned’ datasets, how well does the method perform in realistic, noisy data environments, or when forget/constraint sets are not perfectly curated? Any results or insights into cross-domain/generalization, especially with distributional drift between constraint and attack sets?

- Hyperparameter Sensitivity and Robustness:
How robust is performance (with respect to both constraint satisfaction and model utility) to the choice of $\alpha^k$ and $\eta^k$? Is the anytime constraint guarantee still approximately realized in practice under batch SGD and noisy gradients? Would using adaptive or learned schedules for these parameters improve results?

- Scalability to Larger Models and Real-World Workloads:
Practical impact on compute, memory, and training time when scaling to billion-param models or production settings, where constraints may involve far larger and more diverse safety sets. Any insights or measured overheads?

- Utility-Safety Trade-off Quantification:
Is there a more principled or quantitative way to control or visualize the trade-off between constraint tightness (e.g., lower $\varepsilon$ or larger constraint dataset) and model utility on retained tasks? Could this be integrated into hyperparameter selection or automated tuning?

- Attack Adaptivity and Constraint Bypass:
Have the authors considered more sophisticated adversarial attacks aimed at bypassing explicit constraints (e.g., using paraphrases, alternative prompts, or multi-turn generation) rather than direct fine-tuning on the forget set? How would the current approach handle such cases?

- Open-Set Constraint Handling: What guidance, if any, can the authors provide for constructing constraint datasets in settings where the scope of unwanted knowledge is ambiguous or not fully enumerated? Any ideas for learning constraints or constraint sets adaptively?

---

> ### Author Response · Authors · 2025-12-01
> **Rebuttal to Weaknesses**
>
> Thank you for your review and positive assessment of our work. We appreciate your recognition of the **principled constrained optimization framework, the algorithmic rigor, and the empirical results** demonstrating effective protection against relearning attacks. In this rebuttal, we address your concerns and provide additional clarification to strengthen the manuscript.
>
> For ease of cross-referencing, we number the weaknesses and reply to each accordingly.
>
> 1. We address this concern in our global comments. Please see there for more experiments.
>
> 2. We would like to clarify that RWKU proposes a different evaluation methodology that does not align with our threat model. The established unlearning literature primarily uses datasets such as TOFU, MUSE, and WMDP, which provide well-defined forget sets consistent with our setting. RWKU is posed more towards looking differently at unlearning and how we evaluate it.
> Our evaluation covers the standard benchmarks (TOFU, MUSE) widely adopted in the unlearning community.
>
>     Similarly, "Are We Really Unlearning?" by Hsu et al. (2025) addresses broader conceptual questions about unlearning paradigms rather than proposing specific defensive methods.
>
> 3. To the best of our knowledge, many modern LLMs, including Llama-based models, utilize differentiable activation functions (such as GELU) and all operations in the network are differentiable and smooth (after tokenization). Therefore, the differentiability assumptions are naturally satisfied for these architectures.
>
>     Regarding the hyperparameter choices: Theorem 3.1 establishes a theoretical upper bound on $\eta_k$ based on the problem structure. For $\alpha_k$, our framework guarantees convergence for any value satisfying $\alpha_k \leq \frac{1}{\eta_k}$. As explained in the paper, larger values of $\alpha$ enable more aggressive exploration of the parameter space and allow the algorithm to approach the constraint boundary more closely. The specific choice of $\alpha_k = \frac{1}{\eta_k}$ represents the most aggressive setting allowed by our theory and was validated empirically to provide strong performance. In practice, practitioners can tune $\alpha_k$ within the valid range based on their specific risk tolerance and performance requirements.
>
> 4. We clarify the metrics of the experiments. The 'model utility' metric is not directly optimized by either objective function ($f$ or $g$) in our framework. Rather, it measures broader model capabilities that can be indirectly affected through phenomena such as catastrophic forgetting during fine-tuning. The more pronounced effects in the 8B model reflect the increased difficulty of balancing objectives at larger scales. When using $D_\mathrm{fgt}$ for the main objective, the objectives $f$ and $g$ are fundamentally at odds: $f$ attempts to improve performance on $D_\mathrm{fgt}$, while $g$ explicitly prevents this. Unfortunately, due to resource constraints, we were unable to rerun the experiments for the 8B model with modified hyperparameters or regularizations that could prevent the undesired collapse.
>
>     Notwithstanding, even in this scenario, our framework achieves its primary goal: preventing the relearning of harmful information during an adversarial attempt at fine-tuning. The adversary receives a model with deteriorated 'model utility' scores, but this metric captures only certain aspects of model usefulness and does not reflect safety properties. From a defensive perspective, preventing knowledge restoration while degrading some general capabilities represents a successful outcome—the adversary fails to weaponize the fine-tuning process.
>
> 5. We address this concern in our global comment. Please see therein for more experiments.
>
> 6. We reiterate our response to point 2. Our work does not propose a novel unlearning loss or paradigm shift in unlearning methodology. Rather, our contribution focuses on security-constrained fine-tuning in general and provides a practical algorithm to solve such problems. The mentioned works address different research questions and are not directly applicable to our problem setting.
>
> 7. We will further clarify missing points in the algorithm.
>
> 8. This concern raises an important question that requires case-by-case analysis depending on the specific datasets involved. Our ablation study provides initial insights into this issue by examining scenarios where the constraint uses only a subset of the forget set. In such cases, the adversary effectively introduces data points unknown to the fine-tuner (API provider), which mirrors the reviewer's concern. These experiments demonstrate that our framework maintains robustness even when the adversarial dataset partially differs from the constraint dataset, suggesting resilience to distribution shift between the two.
>
> Please see the global comments for further experiments that address your other concerns.

---

### Official Review · Reviewer_mwMU · 2025-11-03

**Soundness:** 2
**Presentation:** 2
**Contribution:** 1
**Rating:** 2
**Confidence:** 3

**Summary:**

the work proposes a new finetuning method that should increase a model's robustness to finetuning attacks. to achieve this, the authors propose a constrained gradient-descent algorithm, which is evaluated with Llama 3 on the TOFU dataset.

**Strengths:**

- Well motivated problem setting and derivation of constraints and objectives

**Weaknesses:**

- Experiments only on outdated Llama models
- No baselines considered
- Evaluation only on a single dataset? (TOFU)

**Questions:**

see above

---

> ### Author Response · Authors · 2025-12-01
>
> We appreciate the opportunity to address your concerns.
>
> Our experiments utilize Llama 3.1 and 3.2 (released in July and September 2024, respectively), which remain current standards in both academic and industrial settings. While we acknowledge the rapid pace of development in LLMs, our models are not outdated, and our framework is applicable to both modern and older architectures. This work establishes the viability and practicality of our framework, and practitioners can utilize the established framework on their LLM.
>
> Regarding the concerns about baselines and datasets, we have provided comprehensive responses in our global comment, including two new experimental setups demonstrating broader applicability. Please refer to those sections for details.

---

### Author Response · Authors · 2025-12-01
**New Experiment Tasks and Benchmarks**

We provide this global comment with new experiments in response to requests from several reviewers. We believe these results demonstrate the broader applicability of our framework and its competitive performance across diverse tasks.

# New Experiment: LLM Unlearning
We conduct experiments on LLM Unlearning for the TOFU dataset and the MUSE-Books dataset.

## TOFU Benchmark
**Experimental Setup** We focus on the retain90/forget10 task, where the objective is to retain knowledge on 90% of authors while unlearning the remaining 10%. We evaluate our method on Llama 3.2 1B and 3B models using the OpenUnlearning repository with full fine-tuning over 10 epochs. Following the PDU framework, we employ cross-entropy loss for the retain objective and logit-flattening loss for the forget objective. The constraint ensures that retention loss does not exceed 0.3 on the TOFU benchmark.

**Evaluation Metrics.** We report two key metrics (higher is better for both): (1) **Model Utility** measures performance on the *retain* set and auxiliary data, quantifying the model's general capability, and (2) **Forget Success** is the harmonic mean of inverted (1-x) ROUGE and probability scores on the *forget* set, measuring unlearning effectiveness.

**Interpreting Results.** Successful unlearning requires simultaneously maintaining high model utility while achieving high forget success. Methods that excel at only one objective are insufficient. We bolden the methods that best achieve this dual objective. Note that the *target* model represents the original model before unlearning, while the *retrained* model serves as an idealized gold standard (trained from scratch without *forget* data). The retrained baseline is impractical for deployment and is included only for reference.

**Results.** We perform full model fine-tuning for fair comparison across all methods.

**Llama 3.2 1B:**

| Method    | Model Utility | Forget Success |
|-----------|---------------|----------------|
| target    | 0.595         | 0.194          |
| retrained | 0.59          | 0.691          |
| GradDiff  | 0.434         | 0.616          |
| DPO       | 0.561         | 0.603          |
| NPO       | 0.475         | 0.672          |
| SimNPO    | 0.596         | 0.248          |
| RMU       | 0.57          | 0.689          |
| **PDU**       | 0.602         | 0.74           |
| **Ours**      | 0.594         | 0.95           |

**Llama 3.2 3B:**

| Method    | Model Utility | Forget Success |
|-----------|---------------|----------------|
| target    | 0.66          | 0.083          |
| retrained | 0.645         | 0.694          |
| GradDiff  | 0.529         | 0.583          |
| DPO       | 0.609         | 0.54           |
| NPO       | 0.514         | 0.676          |
| SimNPO    | 0.653         | 0.196          |
| RMU       | 0.644         | 0.561          |
| **PDU**       | 0.68          | 0.914          |
|     **Ours**     | 0.674         | 0.941          |


Our method consistently achieves top performance across both model scales. Notably, SimNPO fails to perform effective unlearning, as evidenced by its low forget success scores, which explains its relatively high model utility.

## MUSE-Books Benchmark

**Experimental Setup.** We evaluate unlearning on the MUSE-Books subset using Llama 2 7B. This benchmark measures ROUGE scores on both *retain* and *forget* sets.

**Evaluation Metrics.** Higher retain ROUGE scores indicate better preservation of retained knowledge, while lower forget ROUGE scores indicate more effective unlearning. Successful methods must *simultaneously* achieve high retain ROUGE and low forget ROUGE scores.

**Results:**


| Method    | Retain ROUGE | Forget ROUGE |
|-----------|--------------|--------------|
| target    | 0.691        | 0.64         |
| retrained | 0.687        | 0.196        |
| GradDiff  | 0            | 0            |
| NPO       | 0.551        | 0.303        |
| SimNPO    | 0.531        | 0.252        |
| **RMU**       | 0.626        | 0.225        |
| PDU       | 0.413        | 0.001        |
| **Ours**      | 0.594        | 0.007        |

Both RMU and our method represent Pareto-optimal solutions on this benchmark. RMU achieves higher *retain* ROUGE (0.626 vs. 0.594), while our method achieves substantially lower *forget* ROUGE (0.007 vs. 0.225), demonstrating more complete unlearning with a modest trade-off in retention performance.
This demonstrates that our framework offers flexibility in navigating the retain-forget trade-off space.

(1/3)

---

> ### Author Response · Authors · 2025-12-01
> **New Experiment Tasks and Benchmarks**
>
> # Clarification on Threat Model
>
> As explained in the manuscript, we consider API-based fine-tuning endpoints provided by companies such as OpenAI and Google. To clarify the setting, let Alice denote the LLM provider, and Bob denote a user.
>
> Alice operates fine-tuning endpoints where Bob can upload a dataset. Alice then fine-tunes her proprietary LLM on Bob's dataset and provides Bob with access to a new endpoint for the fine-tuned model.
>
> We assume Bob acts adversarially and provides a dataset that is harmful in nature. For instance, Bob's dataset could contain information for conducting cybersecurity attacks, with the goal of leveraging Alice's LLM capabilities to perform novel attacks.
>
> The experiments in our paper follow this threat model. Specifically, we assume Alice has a model that has undergone unlearning. Bob then attempts to provide samples from the unlearned dataset such that, once Alice's LLM is fine-tuned on these samples, it regains knowledge and capabilities related to the unlearned content.
>
> The following section presents a new experiment setup that also operates under this threat model.
>
> # New Experiment: Aligned Fine-tuning
>
> To further demonstrate the applicability of our framework and algorithm, we consider an aligned fine-tuning setup. Due to computational constraints, we curate a novel experimental setup that effectively showcases the applicability and superiority of our method with our limitations.
>
> **Datasets.** We utilize the *PKU-Alignment/BeaverTails* and *LibrAI/do-not-answer* datasets. Both are question-answering datasets where fine-tuning is performed on the completions to the questions. The adversarial attacker's dataset uses the *30k_train* subset of BeaverTails, from which we randomly select 10K samples marked as unsafe. For the safety constraint dataset, we use the *Claude_response*s from all samples in *LibrAI/do-not-answer* where the response is not harmful (as indicated by the *Claude_harmful* boolean). For the choice of the constraint loss, we employ the length-normalized negative cross-entropy loss.
>
> **Evaluation Metrics.** After fine-tuning, we evaluate three metrics:
>
> 1. **ROUGE-L Recall on BeaverTails *30k_test*:** Measures how much unsafe content the model reproduces (lower is better).
> 2. **ROUGE-L Recall on secondary dataset:** Measures retention of safe responses (higher is better).
> 3. **Toxicity Rate via OpenAI Moderation API:** Evaluates generated content toxicity (lower is better). We generate responses to 1,000 randomly sampled prompts from *bench-llms/or-bench-toxic-all* in the following categories: *self-harm, sexual, harmful, hateful, harassment, violent*. These categories align with our training datasets and facilitate evaluation.
>
> We compare three methods:
>
> 1. **No Defense:** The model is fine-tuned only on the adversarial dataset without using the secondary dataset.
> 2. **Regularization:** A single loss function $L(\theta) = f(\theta) + g(\theta)$ combines both objectives.
> 3. **Our Method:** Employs our constrained optimization framework.
>
> **Results.** We conduct this experiment on Llama 3.2 1B. The results are as follows:
>
> | Method                   | ROUGE-L BeaverTails | ROUGE-L DoNotAnswer | Toxicity |
> |--------------------------|-------------------|-------------------|---------------|
> | No Defense               | 0.366             | 0.264             | 0.440         |
> | Regularization  | 0.37              | 0.89              | 0.36          |
> | Ours                     | 0.35              | 0.92              | 0.27          |
>
> As can be seen, our method provides the best result.
>
> We note that this is a simplified experiment, and more sophisticated setups with different constraint formulations are possible. For example, we can directly incorporate a reward model into the constraint to create an RLHF-like framework, where PPO-based algorithms can be employed to compute gradients and update the model.
>
>
>
> (2/3)

---

> > ### Author Response · Authors · 2025-12-01
> > **New Experiment Tasks and Benchmarks**
> >
> > # Added Baseline for Relearning Defense
> >
> > As requested by reviewers, we add a regularization baseline to the experiments conducted in the paper, which uses a linear combination of the objective and constraint. For ease of comparison, we provide the complete tables below and have updated the corresponding tables in the manuscript. Due to computational constraints, we were able to conduct experiments on the 1B and 3B models at this time.
> >
> > **Llama 3.2 1B**
> > | Dataset | Method         | ROUGE train | ROUGE eval | Forgetting | Utility |
> > |---------|----------------|-------------|------------|------------|---------|
> > |    | naive          | 0.9198      | 0.6978     | 0          | 0.443   |
> > |  $D_{\mathrm{nor}}$        | Reg | 0.8847      | 0.6939     | 0          | 0.3753  |
> > |         | Ours           | 0.9049      | 0.7002     | 0          | **0.4427**  |
> > |||
> > |     | naive          | 0.8905      | 0.7001     | 0.8371     | 0.4947  |
> > |     $D_{\mathrm{NF}}$     | Reg | 0.8278      | 0.6983     | 0          | 0.3972  |
> > |         | Ours           | 0.7617      | 0.6924     | 0.0003     | **0.4561**  |
> > |||
> > |     | naive          | 0.5838      | 0.5943     | 0.6842     | 0.5412  |
> > |     $D_{\mathrm{fgt}}$    | Reg | 0.5792      | 0.5621     | 0          | 0.5593  |
> > |         | Ours           | 0.5477      | 0.5808     | 0.0001     | 0.5652  |
> >
> > **Llama  3.2  3B**
> > | Dataset | Method         | ROUGE train | ROUGE eval | Forgetting | Utility |
> > |---------|----------------|-------------|------------|------------|---------|
> > |    | naive          | 0.91        | 0.7087     | 0.0045     | 0.4768  |
> > |    $D_{\mathrm{nor}}$     | Reg | 0.9158      | 0.7048     | 0          | 0.4446  |
> > |         | Ours           | 0.9381      | 0.7081     | 0          | **0.4756**  |
> > |||
> > |     | naive          | 0.9506      | 0.707      | 0.9086     | 0.5607  |
> > |$D_{\mathrm{NF}}$| Reg | 0.9079      | 0.7087     | 0          | 0.4583    |
> > |         | Ours           | 0.773       | 0.6935     | 0.0004     | **0.5145**  |
> > |||
> > |    | naive          | 0.6156      | 0.5927     | 0.7548     | 0.6001  |
> > |     $D_{\mathrm{fgt}}$    | Reg | 0.3345      | 0.3256     | 0          | 0.4583  |
> > |         | Ours           | 0.6026      | 0.5823     | 0.0001     | **0.6259**  |
> >
> > Our method consistently outperforms the regularization baseline across multiple metrics. We observe that both defense methods successfully prevent knowledge restoration on the forget dataset, as evidenced by *Forgetting* scores near 0 in all cases. However, our method demonstrates superior performance when considering other evaluation metrics.
> >
> > Notably, both methods achieve similar values on the ROUGE-L Val metric, but our method significantly outperforms regularization on the model utility metric. This distinction is particularly important because model utility (as described in the paper, computed as an average of diverse metrics) is not directly optimized by either the primary or secondary datasets. The degraded model utility under regularization provides initial evidence of the *catastrophic forgetting* phenomenon commonly observed in continual learning. This suggests that regularization affects the model's broader capabilities more severely.
> >
> > In contrast, our method operates more surgically—it only activates defensive gradient updates when necessary, thereby better preserving the model's general capabilities while still providing effective defense against adversarial fine-tuning.
> >
> > (3/3)

---

### Meta-Review · Area_Chair_aqYY · 2025-12-19

**Summary:**

Several reviewers viewed the core algorithmic ideas as closely related to existing constrained gradient flow and control-barrier frameworks, raising questions about incremental contribution. Concerns also centered on whether the relearning-attack setting meaningfully advances unlearning or safety research, as opposed to addressing a narrow or artificial scenario.

Despite additional experiments and clarifications in the rebuttal, reviewers did not converge on the work meeting the novelty, scope, and empirical depth expected for ICLR acceptance.

**Reviewer Concerns:**

The rebuttal addressed some surface-level issues, including adding a regularization baseline, expanding experiments beyond TOFU, and clarifying the defender–attacker roles in the threat model.

However, several substantive concerns remain outstanding. In particular, common doubts persist about the conceptual novelty, the limited and still selective experimental coverage (especially the absence of strong, recent unlearning or safety baselines), and the underdeveloped analysis of utility–safety trade-offs and scalability.

Additionally, questions about the practical relevance and realism of the relearning-attack scenario, as well as incomplete positioning with respect to recent unlearning benchmarks and literature, were only partially mitigated and not fully resolved.

**Reviewer Scores:**

The score distribution would most likely remain centered around rejection, with limited upward movement insufficient to change the final recommendation.

---

### Decision · Program_Chairs · 2026-01-26

Reject